# The Resolution of Obstructive Sleep Apnea in a Patient with Goiter after Total Thyroidectomy: A Case Report

**DOI:** 10.3390/reports7020029

**Published:** 2024-04-22

**Authors:** Yacine Ouahchi, Maha Mejbri, Azza Mediouni, Abir Hedhli, Ines Ouahchi, Mounira El Euch, Sonia Toujani, Besma Dhahri

**Affiliations:** 1Pneumology Department/LR 18SP02, La Rabta Hospital, Faculty of Medicine of Tunis, University of Tunis El Manar, Tunis 1007, Tunisia; abirhedhli@hotmail.fr (A.H.); toujanisonia@gmail.com (S.T.); ourari_besma@yahoo.fr (B.D.); 2ENT and Maxillofacial Department, La Rabta Hospital, Faculty of Medicine of Tunis, University of Tunis El Manar, Tunis 1007, Tunisia; maha25841@gmail.com (M.M.); azza.mediouni@gmail.com (A.M.); 3Cytogenetic and Reproductive Biology Department, Farhat Hached University Teaching Hospital, Sousse 4000, Tunisia; ouahchiines@yahoo.fr; 4Internal Medicine Department A/LR 00SP01, Charles Nicolle Hospital, Faculty of Medicine of Tunis, University of Tunis El Manar, Tunis 1007, Tunisia; mouniraach@gmail.com

**Keywords:** obstructive sleep apnea, goiter, thyroidectomy

## Abstract

Obstructive sleep apnea (OSA) may be linked to various factors that narrow the upper airways, such as obesity, adenotonsillar hypertrophy and craniofacial abnormalities. Hypothyroidism has also been described as a risk factor for OSA. However, the implication of goiter independently of thyroid function in the occurrence of OSA remains unclear. We present the case of a 66-year-old woman with a large compressive multinodular goiter for whom total thyroidectomy was indicated. During the preoperative assessment, the patient had a body mass index (BMI) of 37.8 kg/m^2^ with symptoms of OSA. Respiratory polygraphy confirmed the diagnosis of severe OSA (apnea–hypopnea index (AHI) = 32), and treatment with continuous positive airway pressure (CPAP) was initiated prior to thyroid surgery. Surprisingly, after total thyroidectomy, OSA symptoms disappeared, and the patient abandoned CPAP therapy. Subsequent respiratory polygraphy after thyroidectomy showed a decrease in AHI to a normal value (AHI < 5). Interestingly, there was no change in BMI or other factors explaining the resolution of OSA, except for thyroidectomy. This case report suggests that goiter can be considered a risk factor for OSA. However, prospective studies are needed to accurately assess the effects of goiter on the occurrence of OSA according to its dimensions and local extension.

## 1. Introduction

Obstructive sleep apnea (OSA) is characterized by repeated episodes of complete or partial occlusion of the upper airways during sleep [1,2,3]. Its prevalence ranges from 9 to 38% in the general population [4]. Obesity is a major risk factor for OSA, partially explaining its increased prevalence in our society [1,2,3]. Indeed, the accumulation of fat tissue around the upper airways increases their collapsibility and contributes to the severity of OSA [5]. Other risk factors for OSA include age, male gender, tonsillar hypertrophy, and craniofacial abnormalities that narrow the upper airway [1,2,3].

Hypothyroidism has been associated with OSA, mainly due to upper airway narrowing caused by the deposition of mucopolysaccharides and protein extravasation into the tissues of the face, tongue and pharyngeal structures [6,7,8]. The implication of goiter independent of thyroid function in the occurrence of OSA remains unclear [9,10,11,12,13]. Small case series have reported improvements in OSA symptoms after thyroidectomy in patients with goiter [6,7,8,9,14,15,16,17]. However, their findings need validation, and further observations are required to determine if goiter can be considered a risk factor for OSA [9]. Below, we present a case of a patient with a large and compressive multinodular goiter who was diagnosed with OSA before thyroid surgery and whose OSA resolved after total thyroidectomy.

## 2. Case Description

A 66-year-old woman with a goiter and scheduled for total thyroidectomy was referred to the Pneumology Department at La Rabta University Hospital in Tunis, Tunisia, for preoperative respiratory assessment due to suspected OSA.

The patient had an eight-year history of hyperthyroidism associated with multinodular goiter. She was treated with Thiamazole and Propranolol. Additionally, she had comorbidities, including obesity, hypertension, dyslipidemia and type II diabetes, with moderate chronic renal failure. The patient also had a history of lower limb trauma surgery, resulting in limited physical activity.

Total thyroidectomy had recently been indicated due to the progression of the goiter and its compressive nature. The patient achieved a euthyroid state prior to surgery with medication. However, during the preoperative anesthetic evaluation, there was suspicion of OSA, indicated by a STOP-BANG score of 6. Consequently, a specialized respiratory assessment was deemed necessary before proceeding with thyroid surgery.

During the respiratory assessment, no family history or lifestyle habits, such as alcohol consumption and smoking, were identified as risk factors for OSA. However, the patient’s limited physical activity was noted, which could exacerbate obesity, a recognized risk factor for OSA. The patient indicated that she had never consulted a doctor previously for symptoms of OSA or other sleep disorders. However, upon further questioning, the patient reported experiencing snoring without choking or gasping during sleep, as well as excessive daytime sleepiness (Epworth sleepiness scale score: 21/24). Additionally, she complained of frequent nighttime urination (3–4 times per night), non-restorative sleep, morning fatigue and morning headaches. It was considered that hypertension, as a comorbidity, along with nocturia and morning headaches, strongly suggested OSA. Regarding the goiter, it was accompanied by progressive dysphagia and discomfort in the neck without experiencing dyspnea during the day.

The physical examination revealed that the patient measured 149 cm in height and weighed 84 kg, resulting in a BMI of 37.8 kg/m^2^ (classified as obesity: BMI ≥ 30). Her neck circumference was measured at 49 cm in the midway of the neck, between the mid-cervical spine and mid-anterior neck. Upper airway examination showed a Mallampati score of 4, and a large goiter was clinically evident, with no craniofacial abnormalities observed. The lower edge of the thyroid gland was not palpable during the examination. However, a computed tomography (CT) scan of the neck and thoracic inlet confirmed a multinodular enlargement of the thyroid gland, with retro-laryngopharyngeal extension, substernal enlargement and tracheal compression without significant dislocation (Figure 1 and Figure 2).

A home respiratory polygraphy corresponding to a level IIl sleep study was performed. The monitored parameters during sleep included nasal airflow, thoracic and respiratory efforts, oxygen saturation (SpO_2_), heart rate, snoring and body position. Interpretation of the respiratory polygraphy showed severe OSA, with an AHI of 32 events per hour (classified as severe OSA: AHI ≥ 30), primarily characterized by obstructive hypopneas (hypopnea index: HI = 30 events per hour). The oxygen desaturation index (ODI) closely correlated with the AHI, reaching 31 events per hour, but it did not significantly affect the mean nocturnal oxygen saturation (mean SpO_2_) or the cumulative time spent with SpO_2_ below 90% (T90) (mean SpO_2_ = 95%; T90 = 0.4%). The snoring index was high (SI = 65 events per hour), and the most preferred sleep positions were the left side (67% of sleep time) and the right side (25% of sleep time). The supine position, known to favor the occurrence of obstructive respiratory events during sleep, represented only 4% of total sleep time.

Consequently, the patient underwent CPAP titration after reviewing data from an ear, nose and throat (ENT) examination, which revealed no abnormalities that might compromise CPAP therapy. The titration process demonstrated the elimination of abnormal respiratory events with a constant pressure of 7.6 cm H_2_O. Subsequently, CPAP therapy was continued using a nasal mask, resulting in the improvement of symptoms associated with OSA. Good adherence to CPAP treatment was confirmed by a CPAP compliance report, which also indicated the absence of unintentional leaks and a low residual AHI of three events per hour.

After preoperative respiratory management, the patient underwent a new ENT examination with upper airway endoscopy, which revealed the compression of the arytenoids with edema and normal mobility of the two vocal folds. Subsequently, a total thyroidectomy under general anesthesia was performed at the Oto-Rhino-Laryngology Department of La Rabta University Hospital in Tunis, Tunisia. Postoperatively, the patient reported minor changes in voice quality. Endoscopic examination of the upper respiratory tract confirmed transient paresis of the right vocal fold, likely resulting from damage to the innervation provided by the recurrent laryngeal nerve. A macroscopic examination of the resected thyroid gland revealed enlarged dimensions, with the right lobe measuring 7 × 5 × 3 cm, the left lobe measuring 12 × 7 × 5 cm and the isthmus measuring 3 × 2 cm. Histological analysis confirmed the absence of malignancy. The patient received replacement therapy with levothyroxine at a dosage of 125 μg/day.

OSA phenotypes vary significantly when comparing certain anthropometric, clinical and polysomnographic findings [18]. Considering that OSA was primarily related to obesity and advanced age in this case, it was anticipated that the patient would continue to be followed up for OSA and CPAP therapy after total thyroidectomy. However, surprisingly, the patient experienced a complete improvement in OSA symptoms after the thyroid surgery, in addition to the resolution of dysphagia. Consequently, she chose to discontinue her CPAP therapy. Moreover, the patient’s family reported a reduction in snoring following the thyroid surgery, and an assessment of daytime sleepiness using the Epworth sleepiness scale revealed a score of 1/24. Notably, there were no changes observed in the patient’s weight, lifestyle habits or medical conditions or treatments that could explain the resolution of OSA symptoms, apart from thyroidectomy.

Respiratory polygraphy performed without CPAP following total thyroidectomy revealed a significant decrease in the AHI, which reached a normal value (AHI < 5), along with normal levels of nocturnal SpO_2_. These findings strongly suggest the resolution of OSA following thyroid surgery. Additionally, there was a notable decrease in the SI, and the variability in sleep positions was nearly identical to that observed prior to thyroidectomy. The parameters from respiratory polygraphy before and after thyroidectomy are summarized in Table 1.

A medical decision influenced by clinical and polygraphic data was finally made to approve the discontinuation of CPAP therapy for the patient, alongside the implementation of hygienic and dietary measures. However, the patient will undergo follow-up and reevaluation later on. Indeed, her advanced age, obesity and high Mallampati score place her at heightened risk for recurrence of OSA.

## 3. Discussion

It was concluded, in this case, that the large, compressive multinodular goiter was the causal factor for OSA. Indeed, the resolution of OSA after total thyroidectomy was complete, as indicated by clinical and polygraphic data, and unrelated to other factors such as weight loss or specific changes in medical conditions or treatments.

Although a goiter is not typically recognized as a cause of OSA, some case studies have reported improvement in OSA symptoms in patients with a goiter after undergoing thyroidectomy [6,7,8,9,14,15,16,17]. Nevertheless, this improvement has sometimes been assessed solely based on clinical criteria, lacking objective parameters, such as the AHI, to definitively prove the resolution of OSA after thyroidectomy [6,7,8]. Additionally, the degree of improvement in OSA after thyroidectomy has been observed to vary, potentially influenced by factors such as the severity of initial OSA and other contributing factors like advanced age or obesity [6,7,8,9,14,15,16,17]. In this case report, respiratory polygraphy was conducted alongside clinical assessment both before and after thyroidectomy. This approach allowed for the confirmation of OSA diagnosis, the assessment of its severity and the objective evaluation of the patient following thyroid surgery. The reduction in AHI after thyroidectomy supported the resolution of OSA, which appeared complete, leading to the conclusion that the goiter was indeed the causal factor for OSA in this instance. Importantly, no other evident factors were identified to explain the resolution of OSA besides thyroidectomy. 

In this case, a total thyroidectomy was indicated due to the progressive enlargement and compressive nature of the goiter, which resulted in dysphagia and neck discomfort. It also appears that the occurrence and severity of OSA are influenced by the size of goiters and their local extension [7,11]. Indeed, it has been suggested that a large goiter can impede venous return from the head and neck, leading to engorgement and edema of upper airway structures, thereby reducing upper airway patency. Additionally, the displacement of the trachea and other cervical structures associated with a large goiter may interfere with the normal upper airway stiffening reflex that occurs during inspiration. A third hypothesis suggests that a large goiter itself may exert a mass-loading effect on the airways, thereby facilitating the occurrence of abnormal respiratory events during sleep [7].

The clinical and polygraphic data, in this case, reflect certain characteristics of OSA associated with a large and compressive multinodular goiter. These characteristics may be of major interest when studying phenotypes of OSA [18]. Firstly, diurnal and nocturnal symptoms of OSA were identified during the preoperative anesthesia assessment, but they had not previously prompted the patient to seek medical attention. This could be attributed to the gradual progression of the goiter, resulting in the insidious development of OSA symptoms that were underestimated by the patient. Additionally, apart from diurnal symptoms, nocturia and snoring, neither the patient nor her family reported any episodes of breathing cessation. Respiratory polygraphy revealed severe OSA (AHI > 30) with a predominance of hypopneas, indicating that abnormal respiratory events during sleep were primarily associated with episodic partial collapse of the upper airways. Moreover, despite the severity of OSA assessed by AHI, nocturnal SpO_2_ was not significantly disturbed, and there was no requirement for high pressure during CPAP therapy to eliminate abnormal respiratory events.

In this case, no special investigations were performed to diagnose the etiology of OSA or determine the involved pathogenic mechanisms. Initially, it was assumed that OSA was primarily related to obesity and advanced age due to their known effects on upper airway collapsibility [1,2,3]. The resolution of OSA after total thyroidectomy, in this case, led to the conclusion that the goiter was indeed the causal factor for OSA. Studying the behavior of the upper airway under sedation could lead to more precise and personalized approaches for diagnosing and treating respiratory disorders [19]. Drug-induced sleep endoscopy (DISE) allows for the three-dimensional and dynamic visualization of the upper airway during sleep. It represents one of the widespread diagnostic procedures for identifying the anatomical sites of upper airway obstruction in OSA patients, classifying the findings according to the severity and configuration of UA collapse [20,21]. DISE may also play a pivotal role in selecting and customizing candidates for upper airway surgical treatment, as well as mandibular advancement devices (MADs), hypoglossal nerve stimulation and analyzing CPAP failure [20]. In this case, if DISE had been performed, it might have been possible to describe the consequences of goiter on upper airway collapsibility during sleep and to determine the potential implication of other contributing factors for OSA. Indeed, observing dynamics such as the severity of obstruction and patterns of collapse at multiple levels appears to be significant for understanding the underlying causal mechanisms of upper airway obstruction [20].

## 4. Conclusions

This case report demonstrated that a large and compressive goiter constitutes a risk factor for OSA, which can be resolved by total thyroidectomy. Therefore, when evaluating patients with OSA, the thyroid gland should be clinically considered through routine laboratory testing and imaging techniques such as ultrasounds. Moreover, OSA can be suspected in the context of goiter, independent of thyroid function. However, prospective studies utilizing evaluation techniques of upper airways, such as DISE, are needed to accurately assess the effects of goiter on the occurrence of OSA and its characteristics, considering the size and local extension of the enlarged thyroid.

## Figures and Tables

**Figure 1 reports-07-00029-f001:**
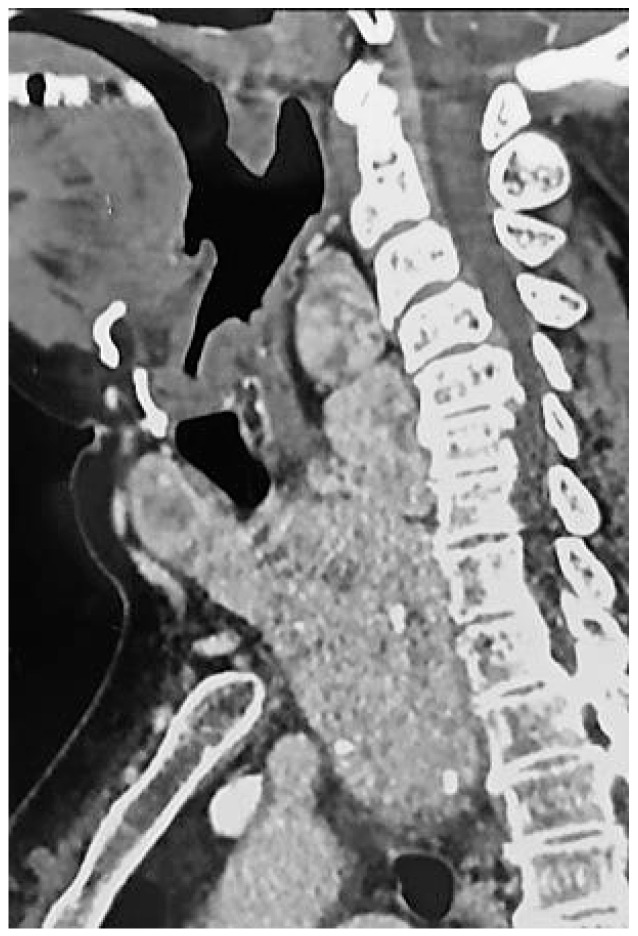
Computed tomography (CT) scan of the neck and thoracic inlet (sagittal view) showing enlargement of the thyroid gland with retro-laryngopharyngeal and retrosternal extensions.

**Figure 2 reports-07-00029-f002:**
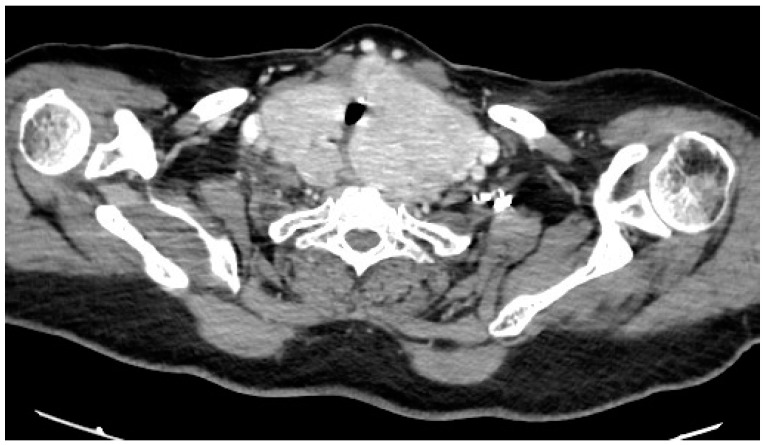
Computed tomography (CT) scan at the level of the thoracic inlet showing enlargement of the thyroid gland with locoregional extension and signs of compression.

**Table 1 reports-07-00029-t001:** Polygraphic parameters before and after total thyroidectomy.

Variables	Before Thyroidectomy	After Thyroidectomy
Evaluation period	22 h 30–05 h 30	23 h 00–07 h 00
AHI, e/h	32 (severe OSA: AHI ≥ 30)	3
AI, e/h	2	0
HI, e/h	30	3
ODI, e/h	31	4
Mean SpO_2_, %	95	95
T90, %	0.4	0
SI, e/h	65	35
Right side, %ST	25	42
Left side, %ST	67	55
Supine, %ST	4	2
Prone, %ST	0	0
Standing, %ST	4	1

AHI—apnea–hypopnea index; AI—apnea index; HI—hypopnea index; ODI—oxygen desaturation index; SpO_2_—peripheral arterial oxyhemoglobin saturation; T90—percentage time of saturation below 90%; SI—snoring index; ST—sleep time; e—event.

## Data Availability

The original contributions presented in the study are included in the article, further inquiries can be directed to the corresponding author.

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
