# Peer review of "The Resolution of Obstructive Sleep Apnea in a Patient with Goiter after Total Thyroidectomy: A Case Report"

_reports, 2024, doi:10.3390/reports7020029_

Round 1

Reviewer 1 Report

Comments and Suggestions for Authors

We have to congratulate the authors for well-written manuscripts but there are serious flaws that should be corrected.

An ENT examination must be performed before any OSA patients should be treated , and before a thyroidectomy please include.

OSA is a disease from the upper airway so a goiter can be a reason for this so it is obvious the relationship. Patients with all the comorbidities refered by authors are  typical from OSA. Not always is suffocation, please refer the importance of headaches, high blood pressure and nocturia as signs of OSA.

Not all the obese patients suffered from OSA, please explain phenotypes.

There is no postoperative image, please include.

The type of polygraphy should be explained.

Reviewer 2 Report

Comments and Suggestions for Authors

This is a very well written report of OSA improvement after thyroidectomy for compressive goiter. However, similar reports have been published, thus the manuscript lacks novelty. I suggest the following in order to improve the scientific value of the manuscript:

-please mention the name of the substance when referring to medication, and not the commercial name (Thyrozol etc)

-please give more details regarding thyroid volume, extension, nodules, etc;

-was the patient dyspneic upon examination or during the day?

-there are some more recent reports regarding the relationship between OSA and goiter/thyroid function (ex: Macias et al Sleep 2020, Shi et al Sleep Medicine 2023) that were not mentioned

- a short literature review comparing the presented cases with other reports would be highly appreciated.

Reviewer 3 Report

Comments and Suggestions for Authors

First, I would like to thank you for reading this manuscript, which presents a case of obstructive sleep apnoea resulting from thyroid goitre. This case presentation is interesting, as both OSA and thyroid goitres are common in everyday practice; therefore, analysing their correlations is interesting and adds to knowledge. However, some issues must be corrected before this manuscript can be considered for publication.

Title

I recommend specifying that this patient underwent a total thyroidectomy; thyroidectomy can refer to other types of thyroid surgery as well.

Abstract

Line 19. It must be clarified if this patient was a man or a woman.

Lines 20-21. What does this BMI category refer to? I.e., specify the BMI category in which this BMI value falls.

Lines 21-22. As in the case of the BMI, the AHI category must also be included. Furthermore, it would be beneficial to specify what symptoms indicated the need for polygraphy.

Introduction

Regarding the risk factors for OSA, considering obesity, I find it essential to mention that the accumulation of adipose tissue near the upper airways is a significant risk factor. Please mention this, including the following reference article: doi:10.3390/life12101543

Lines 41-42. From my point of view, 'have reported improvement in OSA symptoms' would be a better choice.

Case presentation

Line 54. Which type of diabetes was observed?

Lines 62-63. Later in the manuscript, an obese category of BMI was stated; therefore, this patient had another risk factor than thyroid goitres, i.e., obesity. The patient's limited physical activity can be a precipitating factor for fat accumulation. Please clarify this.

Line 72. Please define that this is an obese category of BMI.

Lines 72-73. Where was the neck circumference measured?

Lines 75-77. No trachea dislocation was detected? If not, please state this. It must also be clarified if substernal enlargement was observed.

Line 82. What was the type of polygraph used?

Lines 99-100. If minor changes in voice quality were observed, was a laryngoscopy performed to be sure that a recurrent laryngeal nerve palsy did not occur? Please clarify this.

Line 101. What dosage of L-thyroxine was used following the thyroid surgery?

Lines 102-104. It remains unclear how the other possible causes of OSA were excluded. For this, performing the DISE could have been useful.

Table 1. I recommend including the AHI category before thyroidectomy. Furthermore, please include the 'e' abbreviation in the table caption. 

Regarding the case presentation, it would be interesting to include the result of the histological examination of the removed thyroid glands.

Discussion

Lines 141-143. In addition to the advantages of the applied methods, it must be mentioned that without using the DISE, it is impossible to detect the exact anatomical location of the obstruction.

Line 144. It is unnecessary to reference the tables in the discussion.

Conclusion

Another conclusion of this case presentation can be that, the thyroid gland should also be considered when evaluating patients with OSA (e.g., using routine laboratory testing or thyroid ultrasound should be performed). 

I am looking forward to receiving the revised version of the manuscript.

Comments on the Quality of English Language

Minor editing is necessary. 

Round 2

Reviewer 1 Report

Comments and Suggestions for Authors

Authors have added succesfully the changes suggested by this reviewer.

Reviewer 3 Report

Comments and Suggestions for Authors

Thank you for the revised manuscript. The authors have made efforts to improve the quality of the manuscript, which now reads better. Therefore, from my point of view, it can now be considered for publication. 

Comments on the Quality of English Language

Minor editing is necessary.